# QUANTILE REGRESSION FOR DISTRIBUTIONAL REWARD MODELS IN RLHF

## ABSTRACT

Reinforcement learning from human feedback (RLHF) has become a key method for aligning large language models (LLMs) with human preferences through the use of reward models. However, traditional reward models typically generate point estimates, which oversimplify the diversity and complexity of human values and preferences. In this paper, we introduce Quantile Reward Models (QRMs), a novel approach to reward modeling that learns a distribution over rewards instead of a single scalar value. Our method uses quantile regression to estimate a full, potentially multimodal distribution over preferences, providing a more powerful and nuanced representation of preferences. This distributional approach can better capture the diversity of human values, addresses label noise, and accommodates conflicting preferences by modeling them as distinct modes in the distribution. Our experimental results show that QRM outperforms comparable traditional point-estimate models on RewardBench. Furthermore, we demonstrate that the additional information provided by the distributional estimates can be utilized in downstream applications, such as risk-aware reinforcement learning, resulting in LLM policies that generate fewer extremely negative responses. Our code and model will be released.

## 1 INTRODUCTION

Large Language Models (LLMs) have revolutionized natural language processing, demonstrating remarkable capabilities across a wide range of tasks (Anthropic, 2023; Team et al., 2023; OpenAI, 2023). However, the sheer scale and breadth of their training data present both opportunities and challenges. While LLMs can process and generate human-like text with unprecedented fluency, their outputs may not always align with human preferences, ethics, or real-world applicability. To bridge this gap and ensure that these powerful tools truly benefit humanity, it has been recognized that fine-tuning techniques are necessary that can align LLMs with human values and intentions (Christiano et al., 2017; Ziegler et al., 2019; Bai et al., 2022a). This process of refinement is essential to harness the full potential of LLMs while mitigating potential risks associated with their deployment in real-world scenarios.

Reinforcement learning from human feedback (RLHF) has emerged as a prominent and effective method to align LLMs with human preferences. RLHF uses reinforcement learning (RL) to fine-tune language models by maximizing rewards derived from a trained reward model. This reward model, is itself learned from human preferences. By quantifying human judgments on responses for a prompt, the reward model provides a crucial bridge between human values and the optimization objective of the language model. As a result, accurate reward models are very important in order to finetune LLMs and it has been shown that improvements in reward model quality translates to improvements in the quality of the finetuned LLM (Touvron et al., 2023).

However, current reward models are designed to output a single scalar value for a given query-response pair, an approach that fails to capture the inherent complexity and diversity of human values. This oversimplification can lead to problematic outcomes. For instance, in scenarios where human opinions diverge significantly - with some individuals finding a response appropriate while others deem it inappropriate - the reward model may resort to outputting an average value to minimize penalties across these disparate groups during training. This compromise fails to represent the nuanced spectrum of human preferences accurately. Moreover, it can hinder the learning process

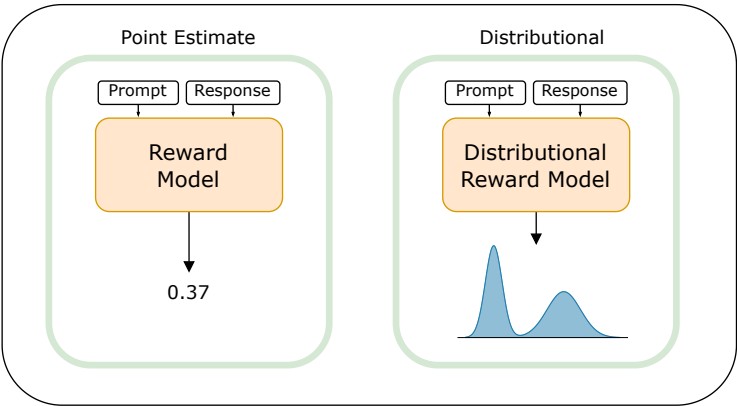

Figure 1: Visualization of two reward estimation methods. The classical point estimate model generates a single scalar value as the reward. In contrast, the distributional reward model proposed in this paper outputs a distribution over possible rewards that can be multimodal.

of the reward model if the training data contains conflicting labels where one kind of response is labeled as preferred in one scenario and as not-preferred in another similar scenario, potentially by different annotators. It has been demonstrated that accurate preferences are crucial for developing robust reward models, with 'incorrect' preferences potentially confusing the model during training. However, in cases of conflicting preferences, the concept of 'correctness' becomes ambiguous, as these divergent views often represent equally valid but different perspectives rather than clear-cut right or wrong answers. This nuance poses a significant challenge to the current paradigm of reward modeling in RLHF.

To address these limitations, we propose a novel approach to reward modeling that reimagines how human preferences are captured and represented. Instead of learning a single scalar reward for each query-response pair, our method, called Quantile Reward Model (QRM), learns a distribution over rewards. Specifically, we employ quantile regression to learn this distribution, allowing us to estimate different percentiles of the reward function and thus capture its entire shape. This probabilistic approach offers a more nuanced and comprehensive representation of human values, capable of capturing the diversity and sometimes conflicting nature of preferences. By modeling a full distribution, we can account for the diverse values and preferences within a population without resorting to oversimplified averages. Furthermore, it addresses the challenge of learning from seemingly conflicting preferences by treating them not as contradictions to be resolved, but as different modes in a multimodal distribution. This shift in perspective allows the model to learn from a wider range of human feedback without the risk of confusion from what might previously have been considered 'incorrect' or conflicting inputs. Consequently, our distribution-based reward model promises to provide a more accurate, flexible, and robust foundation for reinforcement learning in language models, potentially leading to AI systems that are better aligned with the full spectrum of human values.

We implement our approach using a pretrained Llama-3 model Meta (2024) with 8 billion parameters. Our distributional reward model outperforms comparable methods on RewardBench Lambert et al. (2024), a benchmark that evaluates reward models using a point estimate-based metric. This suggests that our model is better at handling conflicting values and the associated label noise during training.

Moreover, the additional information provided by the distributional estimate can be leveraged in downstream applications. We demonstrate this by training an RL policy with a risk-aware utility, derived from the distributional estimates of our model. The resulting policy achieves strong results and generates fewer extreme negative responses compared to a baseline trained using a risk-neutral point estimate as the reward.

## 2 RELATED WORK

RLHF involves using RL to align language models with human preferences or feedback (Christiano et al., 2017; Ziegler et al., 2019; Stiennon et al., 2020). The process generally involves training a reward model on preference data collected from crowdworkers (Bai et al., 2022a; Ouyang et al., 2022) or model-selected responses (Bai et al., 2022b). Once a reward model is developed, RL algorithms can be used to train policies. Another method is to directly optimize a policy by comparing selected and rejected responses using DPO (Rafailov et al., 2023) and followup works (Tang et al., 2024). While this method bypasses the need for an explicit reward model, our focus is on approaches that explicitly train such a model.

Estimating distributions rather than point estimates in regression has a long history and has proven valuable across many fields (Kneib et al., 2023). Quantile regression (Koenker, 2005), in particular, is a powerful tool for approximating distributions. In RL, distributional regression has also demonstrated its effectiveness (Bellemare et al., 2017), with quantile regression achieving especially strong results (Dabney et al., 2018; Kuznetsov et al., 2020; Dorka et al., 2022).

A key distinction exists between epistemic and aleatoric uncertainty (Fox & Ülkümen, 2011; Hüllermeier & Waegeman, 2021). Epistemic uncertainty arises from incomplete knowledge and can be reduced by improving data or models. In contrast, aleatoric uncertainty stems from inherent randomness in the system and cannot be reduced. This paper focuses on modeling aleatoric uncertainty.

Several previous works have modeled preferences using distributions. Distributional Preference Learning (DPL) (Siththaranjan et al., 2023) suggests that hidden contextual factors influence preferences, arising from the diverse values of annotators. This variability is captured by estimating a distribution over rewards. Similarly, the Distributional Preference Reward Model (DPRM) (Li et al., 2024) learns a distribution over rewards, utilizing an optimal transport loss to train the model on preference data. Both DPL and DPRM, like our approach, model a distribution over rewards. However, unlike our method, which incorporates attribute regression, these models rely solely on preference data and optimize the likelihood that the preferred response is ranked higher.

## 3 DISTRIBUTIONAL REWARD MODELS

In this section, we first introduce the reinforcement learning from human feedback (RLHF) framework, then introduce multi-attribute regression reward models, followed by an explanation of quantile regression. Finally, we propose our approach using quantile regression to obtain distributional reward models and explain how it can be used for risk-aware RLHF.

### 3.1 REINFORCEMENT LEARNING FROM HUMAN FEEDBACK

The typical RLHF process using an explicit reward model consists of three stages:

#### 3.1.1 SUPERVISED FINE-TUNING

In the first stage, the pre-trained language model is instruction-tuned by supervised learning on a dataset consisting of a prompt and a high-quality response. Similarly to the pre-training phase the model is trained with a cross-entropy loss over tokens but only on the tokens of the response. The resulting model is often used as the initialization for the reward model and the final policy trained with reinforcement learning.

#### 3.1.2 REWARD MODEL TRAINING

After supervised fine-tuning, the next step is to train a reward model $r_\phi(x, y)$, which evaluates the quality of responses $y$ relative to the prompt $x$ based on human preferences. Typically, reward models are trained on a preference dataset $D_{pref} = \{x, y^-, y^+\}$ consisting of a prompt $x$ and a preferred $y^+$ and not-preferred $y^-$ response. The goal is that the reward model learns to assign higher rewards to outputs that better align with human preferences, such that it can later be used as an informative proxy signal in guiding the optimization of the language model.

The reward model is traditionally trained in accordance with the Bradley-Terry model. The Bradley-Terry model posits that the probability of $x, y^+$ being preferred over $x, y^-$ is given by:

$$P(y^+ \succ y^-) = \frac{\exp(r(y^+))}{\exp(r(y^+)) + \exp(r(y^-))},$$

where we dropped the dependence on the prompt $x$ for notational convenience. The reward model can then be trained to minimize the negative log-likelihood of $y^+$ being preferred over $y^-$:

$$\mathcal{L}_{\text{BT}}(\phi) = -\log P(y^+ \succ y^-),$$

which, after substituting can be written as:

$$\mathcal{L}_{\text{BT}}(\phi) = -r_\phi(y^+) + \log\left(\exp(r_\phi(y^+)) + \exp(r_\phi(y^-))\right).$$

### 3.1.3 REINFORCEMENT LEARNING

Once the reward model is trained, it can be used to optimize the language model further through reinforcement learning. In this stage, the language model generates outputs for a dataset of prompts, and the reward model assigns rewards to these outputs based on their estimated quality. The language model is then trained to maximize these rewards. This iterative process allows the model to improve its performance by continuously refining its behavior according to the feedback provided by the reward model, making it more capable of generating high-quality, human-aligned responses.

More formally, the goal is to finetune the language model $\pi_\theta$ by maximizing the expected reward. To prevent reward hacking, which results in gibberish outputs, the model is penalized for deviating too much a reference policy. This is achieved with a KL divergence penalty between the current policy and the reference policy. Often the initial policy $\pi^{\text{sft}}$ is used as reference policy. The complete RL objective can be expressed as:

$$\mathcal{L}(\theta) = \mathbb{E}_{x \sim \mathcal{D}, y \sim \pi_\theta(y|x)} \left[ r_\phi(x, y) - \beta\, D_{\text{KL}}(\pi_\theta \| \pi_{\text{ref}}) \right],$$

where $\beta$ is a hyperparameter that controls the strength of the KL penalty, balancing the trade-off between maximizing the reward and staying close to the original model distribution. In principle any RL algorithm can be used to optimize the policy. A common choice is to use PPO (Schulman et al., 2017) and more recently more simple REINFORCE based algorithms (Ahmadian et al., 2024).

### 3.2 MULTI-ATTRIBUTE REWARD MODELS

Most existing reward models are trained using the Bradley-Terry loss on binary preference data, where one response is labeled as preferred to another. This approach essentially frames the problem as binary classification. This method, however, fails to account for whether a response was clearly better or only marginally superior to the other. As a result, training a reward model on such data can lead to problems, as it may penalize good responses that are only slightly less favorable than an even better one in the same way it penalizes a very bad response.

Recent datasets are increasingly generated by first collecting absolute labels rather than relative ones (Cui et al., 2023; Wang et al., 2023). Responses are rated across various dimensions such as helpfulness, truthfulness, and harmlessness, with a fine-grained score assigned to each dimension. These individual scores are then aggregated to produce a final score. Preferences are subsequently determined by labeling the response with the higher aggregated score as the preferred one. Alongside these datasets, new approaches have emerged that use regression to directly estimate the fine-grained scores (Wang et al., 2023; 2024a). In this framework, the reward network outputs a single scalar value for each objective and is trained to minimize the mean squared error between its predictions and the fine-grained scores.

For prompts $x$, responses $y$, $M \in \mathbb{N}$ attributes with corresponding scores $l_i$, $i \in [1, ..., M]$ the optimization objective becomes

$$\min_{\theta} \sum_{x,y,l_i \in \mathcal{D}} \sum_{i=1}^{M} ||f_\theta(x,y)_i - l_i||_2^2, \tag{1}$$

where $f_\theta$ is the reward model with $M$ outputs and with parameters $\theta$.

Finally, a single reward score is derived by aggregating the individual scores using a weighted sum. The weights can either be predefined (Wang et al., 2024b) or learned through a gating network. For instance, ArmoRM (Wang et al., 2024a) optimizes the gating network using the Bradley-Terry loss on preference data, where the final rewards are aggregated from individual attribute scores based on the gating weights. During this optimization, the regression network remains frozen. Methods using this approach of first estimating attribute scores have demonstrated significant success, achieving top results on the RewardBench leaderboard (Lambert et al., 2024).

## 3.3 QUANTILE REGRESSION

Quantile regression (Koenker, 2005) is a versatile statistical technique that generalizes the conventional linear regression model by estimating the conditional quantiles of the response variable. Formally, while ordinary least squares (OLS) regression focuses on minimizing the sum of squared residuals to estimate the conditional mean, quantile regression minimizes an asymmetric loss function to estimate specific quantiles belonging to the distribution of the response variable. The quantile function $Q(\tau)$ (where $0 < \tau < 1$) of a probability distribution is the inverse of its cumulative distribution function (CDF) and hence $Q(\tau)$ denotes the value $x$ such that the probability of the corresponding random variable $X$ being lower than $x$ is equal to $\tau$, i.e. $P(X <= x) = \tau$. For a given quantile $\tau$, linear quantile regression solves the following optimization problem:

$$\min_{w} \sum_{i: y_i \geq x_i^\top w} \tau |y_i - x_i^\top w| + \sum_{i: y_i < x_i^\top w} (1 - \tau)|y_i - x_i^\top w|. \tag{2}$$

Here, $y_i \in \mathbb{R}$ represents the labels, $x_i \in \mathbb{R}^d$ the d-dimensional inputs, and $w \in \mathbb{R}^d$ is the vector of coefficients to be estimated.

One of the key strengths of quantile regression lies in its ability to model the conditional distribution of the response variable at different quantiles, offering a more detailed view of the relationship between the dependent and independent variables. This is particularly advantageous in situations where the effects of the input variables are not uniform across the distribution of the outcome variable, such as in the presence of skewed or multimodal distributions, heteroscedasticity, or outliers. Unlike OLS, which provides a single estimate of the central tendency, quantile regression allows to approximately represent a distribution over the response variable, offering more information about the data. Hence, quantile regression provides a comprehensive framework for understanding the distributional effects of input variables on the response variable, making it a critical tool in applications where understanding the full conditional distribution is essential.

## 3.4 DISRTIBUTIONAL REWARD MODELS VIA QUANTILE REGRESSION

Our objective is to develop a reward model that outputs a distribution over rewards. To achieve this, we propose a two-step approach called Quantile Reward Models (QRM). This method involves: (1) estimating distributions over attribute scores (e.g., helpfulness and harmlessness) using quantile regression, and (2) training a gating network to aggregate these individual attribute distributions into a final reward distribution. An illustration of our approach is depicted in Figure 2.

### 3.4.1 STEP 1: ATTRIBUTE DISTRIBUTION ESTIMATION

In the first step, we estimate the distributions for each of the $M$ attributes using quantile regression. Specifically, we perform regression on $K \in \mathbb{N}$ evenly spaced quantiles $\tau_k \in (0, 1)$. For each attribute, we train $K$ linear quantile regression layers, with each layer predicting the value at a specific quantile.

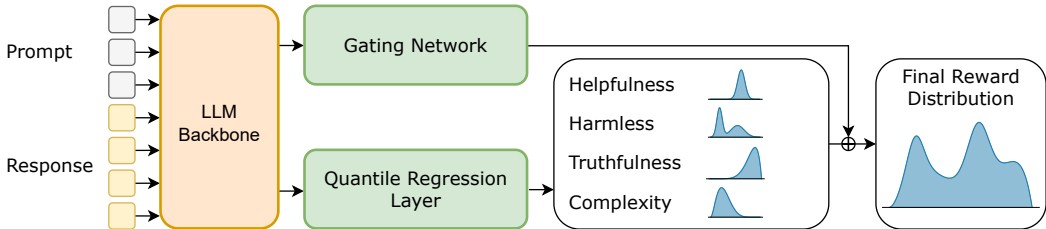

Figure 2: Visualization of our approach. The prompt and response are fed into the LLM backbone, which generates two types of embeddings: a prompt embedding for the gating network and a prompt-response embedding for the quantile regression layers. The quantile regression layers produce quantile estimates for various attributes, such as helpfulness and harmlessness. Simultaneously, the gating network computes weights for the individual distributions corresponding to these quantile estimates. The final output is a mixture distribution, formed by combining the attribute distributions weighted by the outputs of the gating network.

The input to each quantile regression layer is a feature vector derived from a frozen LLM backbone. These feature vectors remain fixed throughout training. This phase requires datasets that provide absolute scores for each attribute to allow for distribution estimation.

### 3.4.2 STEP 2: GATING NETWORK FOR DISTRIBUTION AGGREGATION

In the second step, we train a gating network that predicts the mixing weights for combining the individual attribute distributions into a final reward distribution. The gating network outputs weights $g_m$ for each attribute $m = 1, ..., M$, ensuring that they sum to one: $\sum_m g_m = 1$. These weights determine the contribution of each attribute's distribution to the final distribution. The mixed distribution is then computed as the weighted sum of the $M$ quantile values for each attribute, given by:

$$Q_{\text{mix}}(\tau_i) = \sum_{m=1}^{M} g_m \cdot Q_m(\tau_i). \tag{3}$$

This calculation is performed for all quantiles $\tau_i$. The input to the gating network is a feature vector from the LLM backbone that encodes only the prompt. The gating network is trained using preference data and the Bradley-Terry loss following the approach in (Wang et al., 2024a). To adapt this approach to our scenario, we compute the expected value of the final distribution, which serves as the reward for optimizing the Bradley-Terry loss. Importantly, only the gating network is trained during this stage, while the quantile regression layers remain fixed.

In the end, we obtain both a distribution over attributes and a final distribution over rewards. Notably, our approach can also accommodate fixed attribute weightings, as has been done in works such as HelpSteer (Wang et al., 2023; 2024b).

### 3.5 RISK-AWARE REINFORCEMENT LEARNING FROM HUMAN FEEDBACK

We believe that distributional reward models offer significant potential by leveraging the additional information contained in the reward distribution for downstream reinforcement learning (RL) tasks. In this section, we demonstrate one possible application of this concept.

When deploying a chat model, a critical concern is avoiding particularly poor responses, as low-quality outputs can negatively impact user experience across various dimensions, such as overall quality and safety. Even occasional bad responses can lead to user dissatisfaction. In these cases, it can be beneficial to optimize a metric that emphasizes penalizing low-quality responses as perceived by users.

A point estimate reward model provides only a single value to assess a response. For example, if a response is perceived as excellent by some users and poor by others, a point estimate model may produce an average score that can lead the LLM to generate such mixed-quality responses. In contrast, a distributional reward model allows us to penalize responses with significant disagreement.

One way to achieve this is by applying a concave utility function over the reward distribution to bias the final score to emphasize lower values, i.e., the left tail of the distribution. We define this utility function as:

$$\text{Utility} = \mathbb{E}_{r \sim \mathcal{P}} \left[ -e^{-\lambda r} \right], \tag{4}$$

where $\lambda$ is a hyperparameter controlling the emphasis on low rewards, and the expectation is taken over the distribution $\mathcal{P}$ of reward values $r$ produced by our distributional reward model. We can approximate this by applying the utility function to each quantile estimate before computing the final expectation.

This utility function penalizes distributions with substantial probability mass on low reward values. For instance, a bimodal distribution with one peak at a low value and another at a high value will have a lower utility than an unimodal distribution centered around the same expected value. Consequently, this approach encourages risk-averse policies that avoid producing highly variable or risky responses.

## 4 EXPERIMENTS

### 4.1 IMPLEMENTATION

Our LLM backbone is based on LLaMA-3 with 8 billion parameters (Meta, 2024), initialized with weights from a reward model trained using the Bradley-Terry loss (Dong et al., 2024). The backbone remains frozen during both stages of training. This allows for significant computational efficiency, as we can precompute the backbone features once and reuse them throughout the subsequent stages, thereby significantly reducing the required computation resources. For multi-attribute regression, we follow prior work and use 19 attributes from Wang et al. (2024a), sourced from eight datasets (details provided in the appendix). To limit the computational requirements, we limit the number of data points per attribute to $60,000$ when training the linear quantile regression layers. The quantile regression is implemented using Scikit-learn Pedregosa et al. (2011), with L1 weight regularization set to $0.003$. For each attribute, we train 19 quantile regression models corresponding to the evenly spaced quantiles: $0.05, 0.10, \ldots, 0.90, 0.95$. Additionally, as in Wang et al. (2024a), we mitigate the correlation between the verbosity attribute and other attributes by applying their penalty scores, and adjusting the quantile estimates accordingly.

The gating network used for aggregation is a multi-layer perceptron with three hidden layers and a softmax activation at the output layer. We train the network for 3 epochs using the AdamW (Loshchilov & Hutter, 2019) optimizer with a learning rate of $0.0003$, batch size of $1024$, and a cosine decay learning rate scheduler. The weight decay is set to $0.001$. The training is performed on data from 10 datasets, with further details outlined in the appendix.

### 4.2 REWARDBENCH RESULTS

To examine the general capabilities of our distributional reward model we evaluate it on Reward-Bench Lambert et al. (2024) which is a benchmark for evaluating reward models. It features diverse tasks across 4 categories: Chat, Chat Hard, Safety, and Reasoning. Each category includes datasets with pairwise preference data (chosen vs. rejected responses), and the final score is a weighted average across categories. Following prior work (Kim et al., 2024b; Wang et al., 2024b) we do not evaluate the fifth category of the benchmark named Prior Sets as there are several flaws with this subset (Wang et al., 2024b, p. 25).

We compare our method to various prior methods: HelpSteer2 model with both the Nemotron-4 340B and the LLama-3 70B base model Wang et al. (2024b), ArmoRM Wang et al. (2024a), FsfairX-LLaMA3-RM-v0.1 Dong et al. (2024) which was trained with the Bradley-Terry loss and is used as the backbone for our approach, using GPT4 as a judge Zheng et al. (2023). The comparison to ArmoRM is the most interesting as ArmoRM uses a similar approach but with a point estimate for every attribute. We show the results of the evaluation in Table 1. The results show that QRM achieves the highest performance among models with the same base model. Most notably, it achieves a slightly higher performance than ArmoRM. We suspect a reason is that our model can better handle conflicting annotations during training without confusing the model. Only Nemotron-4 with a much larger model of $340$ billion parameters achieves a higher performance than our model. However, we

| Method | Base Model | Score | Chat | Chat Hard | Safety | Reasoning |
|--------|-----------|-------|------|-----------|--------|-----------|
| HelpSteer2 RM | Nemotron-4 340B | **92.2** | 95.8 | **87.1** | 91.5 | 93.7 |
| ArmoRM | Llama-3 8B | 90.8 | 96.9 | 76.8 | 92.2 | 97.3 |
| HelpSteer2 RM | Llama-3 70B | 86.3 | 91.3 | 80.3 | **92.8** | 90.7 |
| LLM-as-a-judge | GPT-4 Turbo | 84.2 | 95.3 | 74.3 | 87.2 | 86.9 |
| LLM-as-a-judge | GPT-4o | 83.3 | 96.6 | 70.4 | 86.7 | 84.9 |
| Bradley-Terry | Llama-3 8B | 83.6 | **99.4** | 65.1 | 87.8 | 86.4 |
| **QRM** | Llama-3 8B | **91.2** | 97.2 | 78.5 | 91.1 | **98.0** |

Table 1: Performance comparison on RewardBench. The benchmark consists of four categories. The overall score is computed as the weighted average over the single scores. A higher value is better.

note that the main advantage of our approach does not come when evaluated with a metric based on final point estimate but when the additional information is used in a downstream process.

### 4.3 RISK-AWARE RLHF EXPERIMENTS

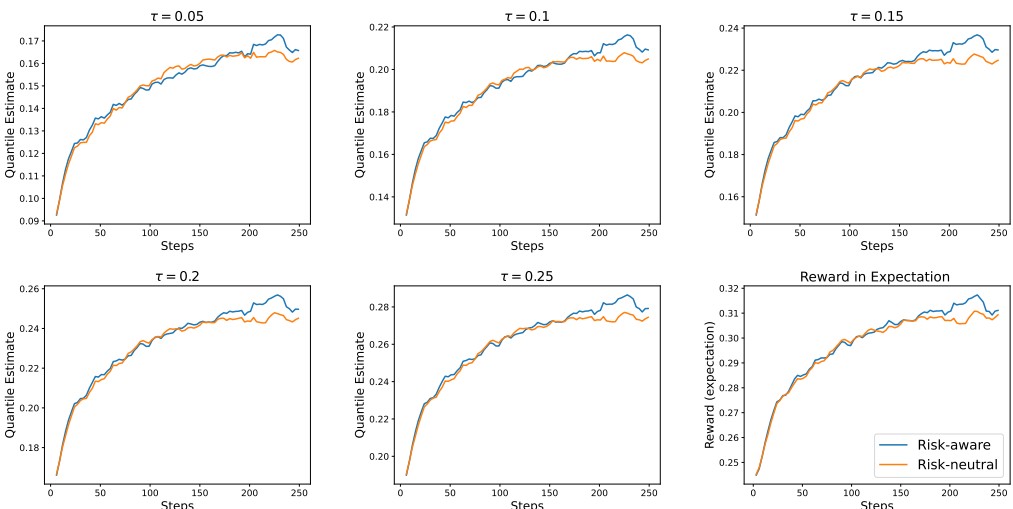

Figure 3: Results for two policies trained with RL using QRM as reward model. One policy was trained with the expectation as reward (risk-neutral) while the other was trained with a risk-aware utility function. We show the quantile estimates for the left tail quantiles $\tau = 0.05, ..., 0.25$ and the expectation of the reward.

To evaluate our approach, we train two RL policies. For comparability, both use our distributional reward model, but in one case we apply the risk-aware utility function to compute the rewards, while in the other, we simulate a point-estimate reward by using the expected value of the reward distribution.

We use the LLaMA-3 model with 8 billion parameters as the base model, training on the Anthropic HH-RLHF dataset (Bai et al., 2022a). Both supervised fine-tuning (SFT) and RL are performed using LoRA, with a rank value of 32 and $\alpha = 64$. For SFT, we train on the responses from the dataset selected as chosen, filtering for examples with at most 1024 tokens. The SFT model is trained for one epoch with a learning rate of 0.0003, batch size of 1024, using the AdamW optimizer and a cosine learning rate schedule.

After completing the SFT stage, we merge the LoRA weights into the model's base weights. For RL, we optimize the LLM policy using the RLOO algorithm (Ahmadian et al., 2024), initializing both the policy and reference policy with the SFT model. We train with a value of $k = 2$ for RLOO,

a fixed learning rate of $0.0001$, and a batch size of $1024$, using AdamW. We filter prompts to those with fewer than $348$ tokens and limit the response length to $512$ tokens.

We present the results of our experiment in Figure 3. The figure illustrates the development of quantile estimates from our distributional reward model for quantiles $\tau = 0.05, \ldots, 0.25$, which represent the left tail of the distribution. The results indicate that the risk-aware utility leads to higher final values for these quantiles compared to the risk-neutral utility. This suggests that the final policy is more risk-aware, resulting in fewer responses that are likely to be deemed very poor or inappropriate. Additionally, we examine the performance of both policies in terms of the expected reward, a risk-neutral metric. Both policies achieve similar final performance on this metric, demonstrating that employing the risk-aware utility does not lead to an overall decrease in performance.

## 5 CONCLUSION

In this work, we proposed QRM, a novel approach for training a distributional reward model in the context of RLHF. Traditionally, reward models are treated as point estimators, which face challenges in accommodating diverse human preferences and handling the resulting label noise during training. In contrast, estimating a distribution that can be multimodal is more powerful, capturing distinct values and preferences more effectively. QRM uses quantile regression to estimate distributions over attribute scores such as helpfulness and harmlessness. A gating network produces weights, aggregating these individual distributions into a final reward distribution.

Our experimental results demonstrate that QRM outperforms previous approaches with comparable base architectures. Notably, even when reduced to point estimates by calculating the expectation of the distribution, our model showed improvements. Moreover, the distributional reward model provides additional information that can be utilized for risk-aware RLHF. The final policy generated by our approach produces responses with higher quantile estimates for the left tail of the distribution, reducing the occurrence of extremely poor responses.

Our work opens several promising avenues for future research. One exciting direction is to directly incorporate conflicting labels into training. Unlike traditional preference learning datasets that rely on majority-vote labels, our model could account for raw, conflicting annotations for the same prompt-response pair. Additionally, it would be interesting to explore integrating our distributional reward model with distributional RL algorithms that estimate a distribution for the value function. Another promising avenue is to leverage the extra information provided by our model within search algorithms during decoding. More broadly, the extra information provided by our distributional reward model allows for more interpretability as well as more creative ways to steer LLMs during downstream fine-tuning.

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

# A    EXPERIMENTAL DETAILS

**Attribute Training Datasets**    To train the quantile regression layers, we follow Wang et al. (2024a) and use the following training datasets with corresponding reward objectives.

- **HelpSteer** (Wang et al., 2023) (35k data): helpsteer-helpfulness, helpsteer-correctness, helpsteer-coherence, helpsteer-complexity, helpsteer-verbosity
- **UltraFeedback** (Cui et al., 2023) (240k data): ultrafeedback-overall-score, ultrafeedback-instruction-following, ultrafeedback-truthfulness, ultrafeedback-honesty, ultrafeedback-helpfulness
- **BeaverTails-30k** (Ji et al., 2023) (30k data): beavertails-is-safe
- **CodeUltraFeedback** (Weyssow et al., 2024) (50k data): code-complexity, code-style, code-explanation, code-instruction-following, code-readability
- **Prometheus** (Kim et al., 2024a) (200k data): prometheus-score
- **Argilla-Capybara**[1] (Daniele & Suphavadeeprasit, 2023) (15k data): argilla-overall-quality
- **Argilla-OpenOrca**[2] (13k data): argilla-judge-lm
- **Argilla-Math-Preference**[3] (2.4k data): This dataset shares the objective ultrafeedback-instruction-following with UltraFeedback

For each objective we normalize the attribute values in the range $[0, 1]$. Further, for each objective we limit the number of data points to $60,000$ for computational efficiency.

**Training Data for the Gating Network**    To train the gating network we again follow the Wang et al. (2024a) and train on the following datasets:

- **HelpSteer** (Wang et al., 2023) (37k pairs)
- **UltraFeedback** (Cui et al., 2023) (340k pairs)
- **SHP** (Ethayarajh et al., 2022) (93k pairs)
- **HH-RLHF** (Bai et al., 2022a; Ganguli et al., 2022) (157k pairs)
- **PKU-SafeRLHF-30K** (Ji et al., 2023)
- **Argilla-Capybara** (15k pairs)
- **Argilla-Math-Preferences** (2.4k pairs)
- **CodeUltraFeedback** (Weyssow et al., 2024) (50k pairs)
- **PRM-Phase-2** (Lightman et al., 2023) (80k pairs)
- **Prometheus2-Preference-Collection** (Kim et al., 2024b) (200k pairs)

---

[1]https://hf.co/datasets/argilla/Capybara-Preferences-Filtered
[2]https://hf.co/datasets/argilla/distilabel-intel-orca-dpo-pairs
[3]https://hf.co/datasets/argilla/distilabel-math-preference-dpo

