# OpenReview forum: "Quantile Regression for Distributional Reward Models in RLHF"
_ICLR.cc/2025/Conference — ICLR 2025 Conference Withdrawn Submission_

### Official Review · Reviewer_a1Bb · 2024-10-28

**Soundness:** 2
**Presentation:** 2
**Contribution:** 2
**Rating:** 3
**Confidence:** 4

**Summary:**

This paper proposes Quantile Reward Models (QRMs) for reinforcement learning from human feedback, offering a distributional approach to reward modeling. Using quantile regression, QRMs estimate a reward distribution rather than a single-point value, aiming to capture human preference diversity and address label noise and preference conflicts.

**Strengths:**

The topic is of interest. The idea of turning the scalar reward into the distributional reward is a trend.

**Weaknesses:**

1.  The idea of replacing scalar reward by distributional reward and the gating mechanism are not brand new ideas \[1,2].&#x20;
2.  The ablation study is not conducted and the QRM ranking is not sufficient enough to validate the effectiveness of the method since the trainset can be varied.
3.  The experiments regarding end-to-end RL process are not sufficient and not informative. The proper baseline RL experiment conducted with normal RM with the same training data is missing.

[1] Wu Y, Sun Z, Yuan H, et al. Self-play preference optimization for language model alignment\[J]. arXiv preprint arXiv:2405.00675, 2024.
[2] Lou X, Yan D, Shen W, et al. Uncertainty-aware reward model: Teaching reward models to know what is unknown\[J]. arXiv preprint arXiv:2410.00847, 2024.

Some other issues:

1.  In Section 3.1.2, the reward model is first referred as $r_{\psi}(x, y)$ (line 157) while in the following text the $\psi$ is missing( line 166).
2.  The structure of the paper needs to be largely improved. For example, Section 3.1 should be renamed as Preliminaries or Backgrounds since it is not your innovation or contributions. The space line before every equations should be removed.
3.  From line 099-103, the citation format should be \citep.
4.  The term <= should be replaced by \leq， which is more formal (line 240)

**Questions:**

Please refer to Weaknesses part

---

### Official Review · Reviewer_DN45 · 2024-11-01

**Soundness:** 2
**Presentation:** 2
**Contribution:** 2
**Rating:** 3
**Confidence:** 4

**Summary:**

This paper proposes QRM to learn preference rewards as distributions via quantile regression. QRM models preference rewards as distributions to handle the diversity and stochasticity within human preference, which are ignored in classic point-wise reward models. The QRM is further used to provide 'risk-aware' rewards for RLHF to emphasize lower rewards.

**Strengths:**

1. Modeling preference rewards as distributions is natural and important, but current SOTA RMs ignore such property of human preference.

2. Quantile regression seems to model reward distributions effectively.

3. Motivation of this paper is well-stated.

**Weaknesses:**

1. Severe lack of empirical study of the proposed method.

2. No ablation study to show the efficacy of each proposed module.

3. Experimental settings are poorly stated, e.g. what model is used to evaluate rewards in fig. 3? how is the rewards evaluated? Is it training rewards or evaluation rewards?

**Questions:**

1. I wonder how important it is to use quantile regression to train QRM? I notice [1] also studies distributional reward models via maximum likelihood estimation. What is QRM's advantage over it?

[1] Lou et al. "Uncertainty-aware Reward Model: Teaching Reward Models to Know What is Unknown"

2. How much performance gain comes from modeling rewards as distributions in QRM? The gating network comes from existing study and I cannot tell the importance of your introduction o quantile regression to reward modeling.

3. How do you evaluate the RLHF results? The only given metric given is the reward curve during training, which is not always in line with LLM performance. Could you provide more convincing results on standard benchmark like AlpacaEval or MMLU to show the effectiveness of risk-aware rewards?

Overall, I feel the empirical and theoretical study of this paper is unable to prove its importance to this community. Although I agree QRM tries to deal with an important topic in LLM alignment, this paper at its current form is not ready for publication.

---

### Official Review · Reviewer_Jpnc · 2024-11-04

**Soundness:** 2
**Presentation:** 1
**Contribution:** 2
**Rating:** 3
**Confidence:** 4

**Summary:**

This paper introduces Quantile Reward Models (QRMs), a novel approach to learn a distribution over rewards rather than a single scalar value in Reinforcement Learning from Human Feedback (RLHF). By moving beyond point estimates, the authors aim to capture a more representative preference distribution, thereby addressing the inherent limitations of scalar-based reward modeling. QRMs employ quantile regression to estimate distributions over attribute scores, followed by training a gating network that assigns weights to the final reward distribution. Experimental results demonstrate that QRMs outperform baseline architectures on RewardBench, particularly in reasoning tasks, and perform competitively in other evaluated tasks.

Additionally, the paper claims that QRMs improve downstream training in RL tasks by introducing a risk-aware utility distribution. This approach biases final scores toward lower values, which penalizes disagreement. The performance of the risk-aware agent was compared to a neutral agent across various values of tau.

**Strengths:**

This research tackles the critical issue of managing disagreement in human feedback—a significant and underexplored direction. The introduction of a quantile approach to model disagreement through distributions, rather than relying solely on scalar values, is a noteworthy innovation. The proposed methodology is compelling in its ability to learn a robust and fair reward distribution by integrating a quantile regression layer with a gating network.

QRMs also contribute to the development of a risk-aware agent, which prioritizes avoiding disagreeable actions and promotes safer
 performance.

**Weaknesses:**

While the methodology is intriguing, several improvements are needed to elevate the paper to a publishable level.

1) Insufficient Performance Analysis: The analysis of QRM’s performance is limited to the primary results in Table 1. Several additional experiments could help substantiate QRM’s effectiveness. For example, what specific training details pertain to the gating network? Does using a selective set of attributes (helpfulness, harmlessness, truthfulness, and complexity) affect performance? Is the reward distribution sensitive to the quality and proportion of disagreeing data? Addressing these questions would make the methodology more convincing and reliable. Additionally, for the results in Table 1, it would be beneficial to explain the standout performance in reasoning tasks. Rather than speculating that the data is more disagreement-sensitive, an analysis with specific metrics on percentage and quality would be more informative.

2) Limited Impact on RL Agent Performance: Although the paper asserts that QRM benefits RL agents, the performance difference in Figure 3 between risk-aware and risk-neutral policies appears minimal. It is standard practice to run training episodes multiple times and present the mean and standard deviation to illustrate differences. The training results also appear unusually similar, possibly due to the same seed? Additionally, adjusting the y-axis scale could highlight performance differences more effectively.

3) Unresolved Philosophical Question Regarding Disagreement: The results prompt an open question regarding policy improvement in the absence of disagreement. Could this lead to more optimal actions, yet inadvertently make the agent overly conservative when confronted with greater diversity in the RL environment? This topic warrants deeper discussion, as the current paper does not fully address this philosophical dimension.

4) Minor Adjustments in Content Balance and Citations: While the first half of the paper devoted extensive space to RLHF basics, the section on related work is relatively sparse. Including recent research on handling noise/disagreement, such as (Jeon et al., 2020) and (Ghosal et al., 2023), would add depth. Additionally, the experiments section could benefit from further expansion.

Jeon, Hong Jun, Smitha Milli, and Anca Dragan. "Reward-rational (implicit) choice: A unifying formalism for reward learning." Advances in Neural Information Processing Systems 33 (2020): 4415-4426.

Ghosal, Gaurav R., et al. "The effect of modeling human rationality level on learning rewards from multiple feedback types." Proceedings of the AAAI Conference on Artificial Intelligence. Vol. 37. No. 5. 2023.

**Questions:**

1) Could you provide additional analysis results, as suggested in Weakness 1, to strengthen the conclusions?

2) Would it be possible to update the results in Figure 3 and the associated discussion to reflect these findings more clearly?

3) Minor request: Could you correct the quotation marks in lines 080 and 095?

**Details Of Ethics Concerns:**

No ethics review.

---

### Official Review · Reviewer_Br2g · 2024-11-04

**Soundness:** 2
**Presentation:** 2
**Contribution:** 2
**Rating:** 5
**Confidence:** 4

**Summary:**

This paper introduces an approach for reward modeling within the context of Reinforcement Learning from Human Feedback. The authors propose Quantile Reward Models (QRMs), which capture a distribution over rewards instead of a single scalar value. This distributional approach is designed to better reflect the diversity and complexity nature of human values and preferences, and accommodate conflicting preferences by modeling them as distinct modes within the distribution. The paper demonstrates that QRMs outperform traditional point-estimate models on the RewardBench benchmark and show the potential of distributional estimates in downstream applications.

**Strengths:**

1. The paper is well-written and easy to follow.
2. This paper tackles a significant challenge by offering a nuanced representation of preferences, which is critical for aligning large language models (LLMs) with human values.

**Weaknesses:**

The results have a few issues which make evaluating the contribution difficult:
1. The results presented in Figure 3 show minimal differences between the risk-neutral and risk-aware policies, making it challenging to assert the superiority of one approach over the other. It would be beneficial if the authors could provide additional metrics or a deeper statistical analysis to highlight the distinctions between the two policies more clearly.
2. The paper presents scores in Table 1 but does not offer a detailed explanation behind the results. For example, it is not clear why the BT model has a higher score in the Chat category or why QRM excels in Reasoning. A more in-depth analysis or additional experiments could provide insights into these performance differences.

While the authors mention training the gating network using preference data and the BT loss, following the approach in Wang et al., 2024a, they do not provide a basic description of the loss form or the training process. This could be confusing for readers who may not be familiar with this previous work. A brief overview would be helpful.

**Questions:**

The motivation and the contribution does not align well. For example, the main motivation of the paper is to better align LLM with human preferences through capturing the diversity and complexity nature of human values and preferences. However, the paper does not include comprehensive end-to-end evaluations to demonstrate how the trained LLM policy achieves better alignment with these values. Instead, the evaluations focus solely on the reward benchmark. Could the authors consider including user studies or LLM evaluations to showcase the practical alignment of their trained models?

---

### Note · Authors · 2024-11-30

**Comment:**

We sincerely thank the reviewers for their thoughtful feedback on our paper. Based on the current reviewer ratings, we understand that the paper is unlikely to be accepted, and we will withdraw our submission. However, we would like to take this opportunity to clarify a few key points about our work that may have been interpreted differently by some reviewers:
- The primary contribution of our paper is modeling the reward as a potentially multi-modal distribution, which sets it apart from prior approaches that use an unimodal probabilistic model for the reward. Our method provides a way to capture and represent conflicting opinions within a population effectively. We further provide a concrete example demonstrating how the additional information from the reward model can be effectively utilized in downstream reinforcement learning fine-tuning.
- The paper by Lou et al., "Uncertainty-aware Reward Model: Teaching Reward Models to Know What is Unknown", referenced in two reviews, was uploaded to arXiv only after the ICLR submission deadline. Furthermore, a significant distinction between their work and ours is that our model explicitly handles distinct modes, as outlined in the first point.

We hope these clarifications offer additional context to better understand and evaluate our work. Once again, we deeply appreciate the reviewers' time and feedback.

**Withdrawal Confirmation:**

I have read and agree with the venue's withdrawal policy on behalf of myself and my co-authors.